# Profiling immuno-metabolic mediators of vitamin B$_{12}$ deficiency among metformin-treated type 2 diabetic patients in Ghana

**Samuel Asamoah Sakyi**[1]*, **Edwin Ferguson Laing**[1], **Richard Mantey**[1],
**Alexander Kwarteng**[2], **Eddie-Williams Owiredu**[1], **Richard Ephraim Dadzie**[3],
**Benjamin Amoani**[4], **Stephen Opoku**[5], **Bright Oppong Afranie**[1], **Daniel Boakye**[6]

1 Department of Molecular Medicine, School of Medicine and Dentistry, Kwame Nkrumah University of Science and Technology, Kumasi, Ghana, 2 Department of Biochemistry and Biotechnology, Kwame Nkrumah University of Science and Technology, Kumasi, Ghana, 3 Department of Medical Laboratory Technology, Faculty of Allied Health Sciences, University of Cape Coast, Cape Coast, Ghana, 4 Department of Biomedical Science, School of Allied Health Sciences, University of Cape Coast, Cape Coast, Ghana, 5 Department of Medical Diagnostics, Faculty of Allied Health Sciences, College of Health Sciences, Kwame Nkrumah University of Science and Technology, Kumasi, Ghana, 6 Department of Epidemiological Methods and Etiological Research, Leibniz Institute for Prevention Research and Epidemiology (BIP-S), Bremen, Germany

* samasamoahsakyi@yahoo.co.uk

**Data Availability Statement:** All relevant data are within the manuscript and its Supporting Information files.

## Abstract

### Background

The association between prolong metformin usage and B12 deficiency has been documented. However, the prevalence estimates of metformin-induced vitamin B12 deficiency showed substantial disparity among studies due to varied study definitions of vitamin B12 deficiency. Metformin blocks the calcium dependent absorption of the vitamin B12-Intrinsic Factor complex at the terminal ileum. Lack of intrinsic factor due to the presence of auto-antibodies to parietal cells (IFA) could lead to vitamin B12 deficiency and subsequently cause peripheral neuropathy. We investigated the prevalence of vitamin B12 deficiency using more sensitive, combined markers of vitamin B12 status (4cB12) and the immuno-bio-chemical mediators of vitamin B12 deficiency.

### Methods

In this observational study, 200 consecutive consenting metformin-treated T2DM patients, aged 35 and above, attending the diabetic clinic at KATH were recruited. Vitamin B$_{12}$ deficiency was classified based on the Fedosov age-normalized wellness quotient. Anthropometric measurement was taken as well as blood samples for immunological and biochemical mediators. Peripheral neuropathy was assessed using the Michigan Neuropathy Screening Instrument (MNSI). Statistical analysis was performed using the R Language for Statistical Computing.

### Results

Using the combined indicator (4cB$_{12}$), the prevalence of metformin induced vitamin B12 deficiency was 40.5% whilst the prevalence of MNSI-Q and MNSI-PE diabetic neuropathy

**Funding:** KNUST/Kref/2018

**Competing interests:** The authors have declared that no competing interests exist.

was 32.5% and 6.5% respectively. Participants with vitamin $B_{12}$ deficiency had significantly higher levels of IFA, GPA, TNF-α, TC, LDL and albumin compared to those with normal vitamin $B_{12}$ levels ($p < 0.05$). Correlation analysis revealed a statistically significant negative association between $4cB_{12}$ and the immunological markers [IFA ($rs = -0.301$, $p<0.0001$), GPA ($rs = -0.244$, $p = 0.001$), TNF-α ($rs = -0.242$, $p = 0.001$) and IL-6 ($rs = -0.145$, $p = 0.041$)]. Likewise, $4cB_{12}$ was negatively associated with TC ($rs = -0.203$, $p = 0.004$) and LDL ($rs = -0.222$, $p = 0.002$) but positively correlated with HDL ($rs = 0.196$, $p = 0.005$).

## Conclusion

Vitamin B12 deficiency and diabetic neuropathy are very high among metformin-treated T2DM patients and it is associated with increased GPA, IFA, TNF-α and cardiometabolic risk factors (higher LDL and TC and lower HDL). Upon verification of these findings in a prospective case-control study, it may be beneficial to include periodic measurement of Vitamin B12 using the more sensitive combined indicators (4cB 12) in the management of patients with T2DM treated with metformin in Ghana.

## Introduction

Prevalence of Diabetes Mellitus (DM) has risen to epidemic proportions in both developed and developing countries. Sub-Saharan Africa is the worse hit with an expected projection of 23.9 million DM cases in the next three decades [1]. Type 2 Diabetes Mellitus (T2DM) is the most abundant form of DM, and accounts for approximately 90% of all DM cases [2]. In Ghana, the disease has affected an estimated 6% of the urban populace [3].

Updated clinical treatment guidelines, including that of the American Diabetes Association, and the European Association for the Study of Diabetes propose that metformin should be initiated with concurrent lifestyle modifications [4–6]. Metformin enhances glucose tolerance in patients with T2DM by lowering basal and postprandial plasma levels of glucose [7]. Serum levels of vitamin $B_{12}$ have been reported to be inversely linked with the duration of T2DM and dose of metformin [1, 3, 8], with an average of 10 to 30% of patients exhibiting malabsorption of vitamin $B_{12}$ [9, 10].

The American Diabetes Association guidelines now recommend periodic evaluation for vitamin B12 deficiency in patients taking metformin [11]. However, measurement of vitamin $B_{12}$ is not done periodically among T2DM patients in Ghana even though a recent study has indicated a high prevalence of metformin-induced vitamin $B_{12}$ deficiency in Ghana [12]. T2DM patients with peripheral neuropathy, who have been treated with metformin for more than 6 months had lower serum vitamin $B_{12}$ and a more severe clinical peripheral neuropathy [13]. This implies that vitamin $B_{12}$ deficiency-induced peripheral neuropathy may be confused with Diabetic Peripheral Neuropathy (DPN) in the absence of the assessment of $B_{12}$ levels [14–16].

Determination of B12 status and results interpretation is not very forthright since different B12 evaluation methods have different specificity and accuracy. Vitamin B12 deficiency is diagnosed by measurements of total serum B12 (sB12), methylmalonic acid (MMA), holo-transcobalamin (holoTC), and total homocysteine (Hcy). The use of individual markers has been demonstrated to be an inadequate reflection of true vitamin $B_{12}$ status. Against this, Fedosov proposed the utilization of combined indicator of vitamin $B_{12}$ status using two or more of the following markers: sB12, MMA, holoTC and Hcy [17]. However, most studies use

only one or two of these biochemical tests in defining vitamin B12 status, leading to contradictory and inconsistent prevalence. It is thus imperative to use combined direct markers of serum vitamin B12 status (sB12 and hTC) and the metabolic markers (MMA and Hcy) ($\omega$; combined indicator of vitamin B12 status using all four indicators (4cB12) in defining vitamin B12 deficiency) [17].

Gastric parietal cells secret intrinsic factor (IF) which plays a vital role in the absorption of vitamin $B_{12}$ in humans. It has been reported that metformin blocks the calcium dependent absorption of the vitamin $B_{12}$-Intrinsic Factor complex at the terminal ileum [18, 19]. Impaired IF production can also occur when there is autoimmune destruction of parietal cells in the context of type 1 diabetes. Lack of IF due to the presence of intrinsic factor antibodies (IFA) could lead to vitamin $B_{12}$ deficiency and subsequently cause peripheral neuropathy, megaloblastic or pernicious anemia [13, 20]. Furthermore, patients with positive anti-gastric parietal cell antibodies (GPA) may present with vitamin $B_{12}$ deficiency. However, in a study by Wang et al [20], only about 12.9% of GPA positive patients actually had pernicious anemia in accordance with the WHO definition. This suggests that a greater proportion of GPA positive cases may have been precipitated by other conditions. Yet, the role of these immunological mediators in vitamin B12 deficiency among metformin treated T2DM patients has not been evaluated. Additionally, $B_{12}$ deficiency has been shown to be linked with elevated levels of pro-inflammatory cytokines (such as tumor necrosis factor-alpha (TNF-$\alpha$) as and biochemical markers related to cardiometabolic risk (glycemia, insulin resistance and lipid profile parameters) in T2DM [21, 22]. Moreover, metformin has been shown to inhibit IL-6 signaling via decreasing IL-6 receptor expression and primary adipocytes cultured in low vitamin B12 conditions showed increased gene expression of pro-inflammatory cytokines including interleukin-6 (IL-6) [23, 24]; however, the relationship is not fully explored in humans.

Thus, studies assessing the relationship between GPA, IFA, TNF-$\alpha$, IL-6 as well as cardiometabolic risk factors and vitamin B12 deficiency among metformin-treated T2DM patients are warranted. Identifying these relationships might elicit specific interventions which may preclude further complications in these patients. This study thus investigated vitamin B12 deficiency using the more sensitive combined indicator of vitamin $B_{12}$ status (4cB12) and evaluated the immune-metabolic mediators associated with $B_{12}$ deficiency among clinically-diagnosed T2DM treated with metformin. The findings from this study will sensitize clinicians to assess metformin-treated T2DM patients for vitamin $B_{12}$ deficiency and related immuno-biochemical disorders to optimize care for these patients.

## Materials and methods

### Study design

This hospital-based observational study was conducted at the diabetic clinic of the Komfo Anokye Teaching Hospital (KATH) in the Ashanti Region of Ghana, from January 2018 to March 2019. KATH has approximately one thousand-bed facility and serves as a main referral point for parts of the northern and middle belts of Ghana. The hospital has a diabetic clinic which is frequented by more than 100 patients per week [25].

### Ethical consideration

This study was approved by the Committee on Human Research Publication and Ethics (CHRPE) of the School of Medicine and Dentistry (SMD), Kwame Nkrumah University of Science and Technology (KNUST) (CHRPE/AP/013/18) and the ethical review board of KATH. Participation was voluntary and written informed consent was obtained from each participant after the aims and objectives of the study had been explained to them.

## Study population

Two hundred (200) patients diagnosed with T2DM attending the diabetic clinic at KATH and who gave informed their consent were enrolled into the study. Since T2DM is predominantly a disease of the old, we included participants who were above 35 years of age.

## Inclusion and exclusion criteria

Clinically diagnosed T2DM patients treated with metformin, aged 35 years and above receiving medical care at the out patients' department of the diabetic unit of KATH were recruited onto the study. The diagnosis of type 2 diabetes mellitus was made based on American Diabetes Association (ADA) criteria (based on defective progressive insulin secretory on the background of insulin resistance using fasting plasma glucose $\geq$7.0 mmol/L; glycated hemoglobin $\geq$6.5%; oral glucose tolerance test 11.1 mmol/L) [26]. Patients with history of anemia, blood transfusion, thyroid illness, chronic alcoholism, renal insufficiency, gastric surgery as well as patients on chronic parenteral or enteral nutritional support, on $B_{12}$ supplementation (parenteral or oral), proton pump inhibitors, malabsorption syndrome, liver disease, vegetarians and pregnant women were excluded from this study. Patients with type 1 diabetes (diagnosed based on high plasma glucose in addition to autoimmune indicators such as autoantibodies to insulin, autoantibodies to glutamate decarboxylase (GAD65)) were excluded.

## Questionnaire administration

Validated structured questionnaires were administered to gather information on demographic characteristics, anthropometric variables, medications history including dosage, duration and symptoms of neuropathy. Patients' medical records were retrieved from the hospital's archive and reviewed to obtain additional relevant information such as medications used.

## Assessment of peripheral neuropathy using the Michigan Neuropathy Screening Instrument (MNSI)

Neuropathy was assessed using the MNSI. The MNSI history questionnaire was self-administered and consists of responses that are collated to obtain a total score as described by Feldman et al. [27]. A score of $\geq 7$ was considered abnormal.

## Physical assessment

In all assessments, we ensured that the foot was warm (>30˚C). During the MNSI assessment, a healthcare professional inspected both feet for malformations, calluses, dry skin, fissures and infections. Other deformities such as overlapping toes, joint subluxation, halux valgus, hammer toes, prominent metatarsal heads, flat feet, amputation and medial convexity (Charcot foot) were also assessed. Any abnormality observed on each foot (including ulcerated foot) received a score of 1.

## Vibration sensation

Vibration sensation was assessed bilaterally using a 128 Hz tuning fork placed over the dorsum of the great toe on the boney prominence of the distal interphalangeal joints (DIP) joint. Patients (with eyes closed) were asked to indicate when they could no longer sense the vibration from the tuning fork. If the examiner felt vibration for 10 or more seconds on his or her finger, then vibration was considered decreased. Vibration was scored as 1) present if the examiner sensed the vibration on his or her finger for < 10 seconds, 2) reduced if sensed for $\geq$ 10 or 3) absent (no vibration detected.)

## Muscle stretch reflexes

The ankle reflexes were examined using an appropriate reflex hammer (e.g. Trommer or Queen square). The ankle reflexes were elicited in a sitting position with the foot when the patient is relaxed. For the reflex, the foot was passively positioned and the foot dorsiflexed slightly to obtain optimal stretch of the muscle. The Achilles tendon was tapped directly. If the reflex was obtained, it was graded as present. If the reflex was absent, the patient was asked to perform the Jendrassic maneuver (i.e., hooking the fingers together and pulling). Reflexes elicited with the Jendrassic maneuver alone were designated "present with reinforcement." If the reflex was absent, even with the Jendrassic maneuver, the reflex was considered absent. The total possible score is 8 points and, by the scoring algorithm we used, a score greater than or equal to 2.5 was considered abnormal [27].

## Blood pressure measurement

Blood pressure was measured using a calibrated sphygmomanometer with appropriate cuff sizes and stethoscope in accordance with the recommendation of the American Heart Association (American Heart Association 2012). The measurement was taken with participants in sitting position and after having rested for at least 10 min. Triplicate readings were taken 5 min apart and the mean value was recorded to the nearest 1.0 mm Hg.

## Measurement of Body Mass Index (BMI)

The height of the participants was measured to the nearest meter using a wall-mounted ruler and weight was measured in light clothing without shoes, and in an upright position (to the nearest 0.1 kg) using a bathroom scale (BR9012, Zhongshan Camry Electronic Co. Ltd, Guangdong, China). BMI was calculated as weight (kg) divided by height squared ($m^2$).

## Blood sample collection and processing

Five milliliters (5ml) of venous blood sample was collected from each participant into both serum gel separator tube (3ml) and EDTA tubes (2ml). EDTA samples were processed for analysis immediately for hemogram and samples in serum gel separator tube were centrifuged at 1500 rpm for 3 min. The serum was stored in cryovials at $-80°C$ until vitamin B12, and methyl malonic acid (MMA) assays were performed.

## Measurement of hematological parameters

Both thick and thin blood films were performed by standard protocols as described by [28]. Full blood count (FBC) was estimated using an automatic hematological analyzer (SYSMEX XP-2000i, Japan).

## Biochemical assays

Serum samples collected from the participants were analyzed for the concentrations of vitamin $B_{12}$, GPA and IFA by solid phase sandwich ELISA using the DuoSet ELISA kit (R&D Systems, Inc., USA) according to the manufacturer's instructions. To improve the diagnosis of vitamin $B_{12}$ deficiency, methylmalonic acid and homocysteine blood levels were also measured, as elevated levels are sensitive indicators of tissue vitamin $B_{12}$ deficiency [29]. These markers of vitamin $B_{12}$ deficiency have been shown to improve the sensitivity and specificity for detecting vitamin $B_{12}$ deficiency especially among type 2 diabetic patients with borderline serum vitamin $B_{12}$ concentrations of 200–400 pg/ml and subtle haematological manifestations [30]. Positive GPA was defined as values >20 pmol/l whereas IFA positivity was also defined by >1.53 AU/mL.

## Classification for vitamin $B_{12}$ deficiency

Vitamin $B_{12}$ deficiency was classified based on the Fedosov age-normalized wellness quotient [ω; combined indicator of vitamin $B_{12}$ status using all four indicators ($4cB_{12}$)] [17]. The wellness score was calculated according the formula: $\omega$ ($4cB_{12}$) = $\log_{10}[(sB_{12}{*}hTC)/(MMA{*}Hcy)]_{test} - [lr_{norm}/(1- (Age/230)^{2.6})]$, where the direct markers of vitamin $B_{12}$ status ($sB_{12}$ and hTC) are reported in pmol/L and the metabolic markers (MMA and Hcy) are reported in μmol/L. The *test* portion of the equation refers to the tests of the participant. The second component of the equation describes the "normal" logarithmic ratio ($lr_{norm}$) predicted based on the age of the participant. The absolute values of the correction factor $lr_{norm}$ decrease with increasing age: for age $\leq$ 40 years, the $lr_{norm}$ = 3.750; 41–60 years (3.678); 61–80 years (3.561) and >80 years (3.436). Participants with $\omega<$ -0.5 were considered vitamin $B_{12}$ deficient in accordance with previous studies [17, 31, 32].

## Statistical analysis

Statistical analysis was performed using the R Language for Statistical Computing version 3.6.0 [33]. Categorical data were presented as frequencies (percentages). For continuous data, normality was checked using Shapiro-Wilk's test, as well as visual inspection with Q-Q plots. Parametric and nonparametric data were presented as mean ±SD and median (interquartile ranges), respectively. Distribution of the indicators of vitamin $B_{12}$ ($4cB_{12}$, $sB_{12}$, hTC, MMA, Hcy) were presented with density plots. Comparison of immuno-metabolic parameters based on vitamin $B_{12}$ status was performed using one-way analysis of covariance (ANCOVA) with adjustment for age, sex, BMI, duration of diabetes, and dosage and duration metformin therapy. Hierarchical clustering by Spearman's correlation was used to assess relationship between the indicators of vitamin $B_{12}$ and immuno-metabolic parameters. Logistic regression analysis was used to evaluate the association between vitamin B12 status and diabetic neuropathy. All tests were two-sided and p-value < 0.05 was considered statistically significant.

## Results

A total of 200 diabetic patients with mean age of 59.00±9.11 years were included in this study. Most of the participants were females (71.0%), have had diabetes for 13.92±6.19 years and had been on metformin therapy for 12.88±6.15 years. Furthermore, 32.5% and 6.5% presented with neuropathy based on MNSI-Q and MNSI-PE, respectively. Other clinical, immunological, metabolic and hematological parameters are shown in **Table 1**.

The average $sB_{12}$, hTC, MMA and Hcy were 142.10 (59.00–178.90) pmol/L, 27.02 (15.03–45.66) pmol/L, 0.17 (0.14–0.20) μmol/L and 6.50 (4.35–13.80) μmol/L, respectively. The median Fedosov's wellness score (ω) was -0.18 (-0.97–0.32) (**Fig 1A**). Using the combined indicator ($4cB_{12}$), the prevalence of vitamin $B_{12}$ deficiency was 40.5% (**Fig 1B**).

IFA (1.18 (1.00–1.30) vs 1.00 (0.98–1.20), AU/ML, p = 0.003), GPA (5.80 (4.10–7.90) vs 4.80 (3.60–6.50) pmol/l, p = 0.005) and TNF-α (45.00 (33.47–56.67) vs 41.67 (28.33–48.33), pg/ml p<0.0001) levels were significantly higher among the participants with vitamin $B_{12}$ deficiency compared to those with normal vitamin $B_{12}$ levels. The levels of IL-6 did not differ significantly between the deficient and non-deficient group (**Fig 2**).

Participants with vitamin $B_{12}$ deficiency presented with significantly higher TC (5.60 (4.60–6.80) vs 5.10 (4.10–6.00) mmol/L, p = 0.01), LDL (3.08 (2.46–4.84) vs 2.86 (2.15–3.85) mmol/L, p = 0.005) and albumin (4.20 (4.10–4.30) vs 4.10 (4.00–4.20) g/l, p = 0.021) levels compared to those with normal vitamin $B_{12}$ levels (**Fig 3**).

**Table 1. Demographic, clinical and immune-metabolic profile of the entire study participants.**

| Variables | Mean ±SD |
|---|---|
| Age (years) | 59.00±9.11 |
| BMI (kg/m$^2$) | 26.10±3.70 |
| SBP (mmHg) | 132.21±17.60 |
| DBP (mmHg) | 77.35±9.93 |
| DM duration (years) | 13.92±6.19 |
| Metformin duration (years) | 12.88±6.15 |
| **Sex** | **Frequency (%)** |
| Female | 142 (71.0) |
| Male | 58 (29.0) |
| **Metformin dosage (mg/day)** | |
| 1000 | 32 (16.0) |
| 2000 | 130 (65.0) |
| 3000 | 38 (19.0) |
| **MNSI-Q** | |
| Neuropathy absent | 135 (67.5) |
| Neuropathy present | 65 (32.5) |
| **MNSI PE** | |
| Neuropathy absent | 187 (93.5) |
| Neuropathy present | 13 (6.5) |
| **Circulation impairment** | |
| Absent | 112 (56.0) |
| Present | 88 (44.0) |
| **General asthenia** | |
| Absent | 110 (55.0) |
| Present | 90 (45.0) |
| **Immunological parameters** | |
| IFA (AU/mL) | 1.10 (1.0–1.21) |
| GPA (pmol/l) | 5.20 (3.71–6.80) |
| TNF-α (pg/ml) | 42.67 (30.00–50.92) |
| IL-6 (pg/ml) | 66.67 (58.17–85.00) |
| **Metabolic parameters** | **Median (IQR)** |
| FPG (mmol/L) | 6.90 (6.00–9.05) |
| HbA1c (%) | 6.90 (6.00–8.20) |
| TC (mmol/L) | 5.10 (4.20–6.30) |
| TG (mmol/L) | 1.60 (1.20–1.90) |
| HDL (mmol/L) | 1.30 (0.90–1.90) |
| LDL (mmol/L) | 3.02 (2.23–4.32) |
| Serum albumin (g/dl) | 4.10 (4.05–4.20) |
| Corrected calcium (mmol/L) | 9.04 (8.82–9.44) |
| **Hematological parameters** | |
| TWBC (10$^3$/$\mu$L) | 5.12±1.71 |
| RBC (10$^6$/$\mu$L) | 4.02±0.60 |
| Hemoglobin (g/dL) | 12.05±1.54 |
| HCT (%) | 34.12±4.79 |
| MCV (fL) | 83.39±7.94 |
| MCH (pg) | 30.82±2.44 |
| MCHC (g/dL) | 36.56±1.87 |

(*Continued*)

**Table 1.** (Continued)

| Variables | Mean ±SD |
| --- | --- |
| PLT ($10^3/\mu L$) | 187.0 (155.0–210.0) |

MNSI-Q: Michigan Neuropathy Screening Instrument Questionnaire; Michigan Neuropathy Screening Instrument Physical Examination (MNSI-PE); IFA: Intrinsic factor antibody; GPA: Gastric parietal cell antibody; TNF-α: Tumor necrosis factor-alpha; IL-6: Interleukin 6; FPG: Fasting plasma glucose; HbA1c: Glycated hemoglobin; TC: Total cholesterol; TG: Triglyceride; HDL: High density lipoprotein; LDL: Low density lipoprotein; TWBC: Total White Blood Cells; RBC: Red Blood Cells; HCT: Haematocrit; MCV: Mean Cell Volume; MCH: Mean Cell Haemoglobin; MCHC: Mean Cell Haemoglobin Concentration; PLT: Platelet count.

Although vitamin B12 deficient participants had marginally higher mean cell volume (macrocytosis) and platelet count, there was no statistically significant association between hematological parameters and vitamin B12 status (S1 Table in S1 File).

Correlation analysis revealed a statistically significant negative association between $4cB_{12}$ and the immunological markers [IFA (rs = -0.301, AU/ml, p<0.0001), GPA (rs = -0.244, pmol/l, p = 0.001), TNF-α (rs = -0.242, pg/ml, p = 0.001) and IL-6 (rs = -0.145, pg/ml, p = 0.041)]. Likewise, $4cB_{12}$ was negatively associated with TC (rs = -0.203 mmol/l, p = 0.004) and LDL (rs = -0.222, mmol/l, p = 0.002) but positively correlated with HDL (rs = 0.196, mmol/l, p = 0.005). A negative correlation was also observed between $sB_{12}$ and IFA (rs = -0.341, p<0.0001), GPA (rs = -0.234, p = 0.001) and TNF-α (rs = -0.169, p = 0.016) whereas hTC correlated negatively with only TNF-α (rs = -0.22, p = 0.002). Conversely, MMA correlated positively with AIF (rs = 0.189, p = 0.007), GPA (rs = 0.262, p<0.0001), IL-6 (rs = 0.185, p = 0.009) and calcium (rs = 0.145, p = 0.041) but negatively with HDL (rs = -0.19, p = 0.007). Correspondingly, Hcy showed a positive correlation with the immunological markers [AIF (rs = 0.278, p<0.0001), GPA (rs = 0.342, p<0.0001), TNF-α (rs = 0.174, p = 0.014), IL-6 (rs = 0.186, p = 0.008) as well as TC (rs = 0.211, p = 0.003) and LDL (rs = 0.217, p = 0.002) but a negative correlation with HDL (rs = -0.18, p = 0.011) (**Fig 4**).

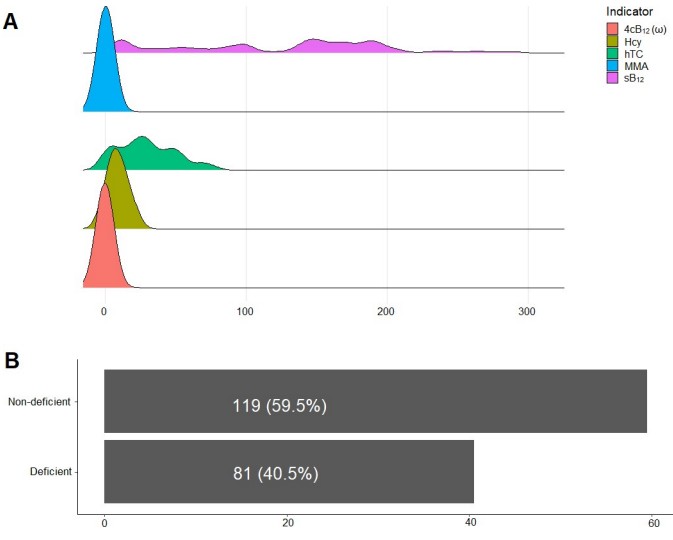

**Fig 1. Distribution of vitamin B12 indicators and the prevalence of B12 deficiency.** (A) Density plots of vitamin B12 indicators; (B) Bar graph displaying the prevalence of vitamin B12 deficiency using 4cB12. 4cB12 (ω): combined vitamin B12 indicator, Hcy: Homocysteine, hTC: holotranscobalamin, MMA-methylmelanoic acid, SB12-Serum vitamin B12.

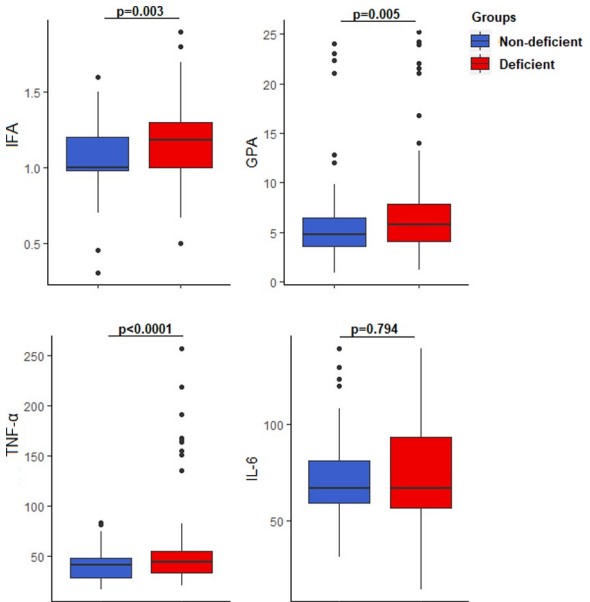

**Fig 2. Comparison of select immunological parameters based on vitamin B$_{12}$ status.**

Among the 32.5% and 6.5% presented with neuropathy based on MNSI-Q and MNSI-PE, respectively, a higher proportion also had vitamin B12 deficiency (20.0% and 4.0% for MNSI-Q and MNIS-PE, respectively). Furthermore, vitamin B12 deficiency was significantly associated with over 3-fold increase in the odds of neuropathy compared to the non-deficient counterparts by MNSI-Q (COR = 3.67, 95% CI (1.97–6.82), p<0.0001). Although vitamin B12 deficiency was also associated with higher odds of neuropathy by MNSI-PE, the association was not statistically significant (**Table 2**). Of note, a higher dose of metformin (3000mg/day) but not duration of treatment was associated with increased odds of vitamin B12 deficiency and neuropathy (S2–S4 Tables in S1 File).

## Discussion

The current study investigated the prevalence of vitamin B12 deficiency among T2DM patients treated with metformin using the more sensitive combined indicator of vitamin B12 status (4cB12) and immuno-biochemical mediators of vitamin B12 deficiency.

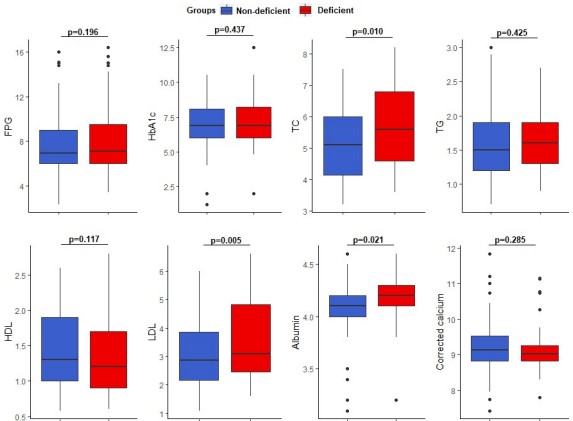

**Fig 3. Comparison of select biochemical parameters based on vitamin B$_{12}$ status.**

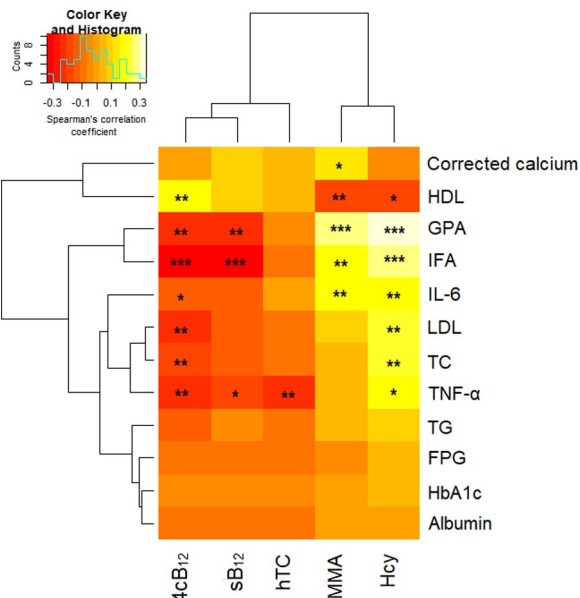

**Fig 4. Correlation between indicators of vitamin B$_{12}$ and the immunological parameters.** Hierarchical clustering by Spearman's correlation was used to assess relationship between adipokines and their ratios with obesity indices. Red-yellow-white coloration represents negative (-) to positive (+) correlation coefficient. *; $p < 0.05$, **; $p < 0.01$, ***; $p < 0.0001$.

## Prevalence of vitamin B12 deficiency

Vitamin B12 deficiency is diagnosed by measurements of total B12, methylmalonic acid (MMA) holo-transcobalamin (holoTC), and total homocysteine (Hcy) in blood. However, most studies use only one or two of these biochemical tests in determining vitamin B12 deficiency and this has often led to inconsistent and contradictory results. The current study however classified Vitamin B12 deficiency using combined direct markers of vitamin B12 status (sB12 and hTC) and the metabolic markers (MMA and Hcy) based on the Fedosov age-normalized wellness quotient [ω; combined indicator of vitamin B12 status using all four indicators (4cB12)] [34]

Using this combined indicator (4cB$_{12}$), the prevalence of vitamin B$_{12}$ deficiency among T2DM patients was 40.5%. Sparre Hermann et al. reported a 26.7% prevalence of vitamin B12 deficiency in metformin-treated T2DM patients based on holoTC in Sweden [35]. Studies by Ahmed et al., in South Africa and Qureshi et al., reported deficiencies of 28.1%, and 33% respectively [10, 36]. A related study in Ghana, by Yakubu et al reported vitamin B12 deficiency of 32.1% using serum vitamin B12, methylmalonic acid as markers [12]. The varied definitions of

**Table 2. Association between vitamin B12 status and diabetic neuropathy.**

| Vitamin B12 status | MNSI-Q | | | p-value |
|---|---|---|---|---|
| | Neuropathy absent | Neuropathy present | COR (95% CI) | |
| Non-deficient | 94 (47.0) | 25 (12.5) | 1 | |
| Deficient | 41 (20.5) | 40 (20.0) | 3.67 (1.97–6.82) | <0.0001 |
| | MNSI-PE | | | |
| Non-deficient | 114 (57.0) | 5 (2.5) | 1 | |
| Deficient | 73 (36.5) | 8 (4.0) | 2.50 (0.79–7.93) | 0.120 |

vitamin B12 deficiency by the use only one or two biochemical indicators of vitamin B12 deficiency these studies could explain these wide variations in the reported prevalence. Moreover, differences in cultural and religious beliefs in different geographic regions of the world, could also affect the observed variations. Of note, the high rate of vitamin B12 deficiency in this study compared to previous studies could be suggestive that the use of individual markers (serum B12, MMA, hTC or Hcy) underestimates the vitamin B12 status among diabetic populations.

## Influence of immunological mediators

The current study observed that intrinsic factor antibody (IFA), gastric parietal cell antibodies (GPA) and tumor necrotic factor alpha (TNF-$\alpha$) levels were significantly higher among vitamin $B_{12}$ deficient T2DM patients compared to those with normal vitamin B12 levels. Antonio Cabrera de León et al in a similar study in the Canary Island estimated a higher prevalence of GPA (7.8%) [30]. Patients with increased positive GPA are susceptible to gastric parietal cell destruction that could lead to a deficiency in IF and subsequent blockage in the absorption of vitamin B12 in the terminal ileum [19, 24] Additionally, vitamin B12 absorption requires IF and auto-immune destruction of IF results in decreased vitamin B12 levels as consistent with this study [13, 20].

Furthermore, low vitamin $B_{12}$ levels has been reported to be associated with increased pro-inflammatory cytokines levels and could account for the high TNF-$\alpha$ levels among vitamin $B_{12}$ deficient participants compared to the non-deficient counterpart [21, 22]. Our study also observed that an increment in GPA, IF, and TNF-a led to the increase in methylmalonic acid (MMA), holo-transcobalamin (holoTC), and total homocysteine (Hcy) (4cB12) [**Fig 4**]. Thus, GPA, IF and TNF-$\alpha$ are independent risk factors associated with vitamin $B_{12}$ deficiency among metformin-treated T2DM.

## Relationship between vitamin $B_{12}$ and biochemical parameters

One major function of Vitamin $B_{12}$ is to act as a coenzyme in the synthesis of succinyl-CoA from methylmalonyl-CoA (MM-CoA) [37]. In the event of vitamin $B_{12}$ deficiency, this reaction is blocked which results in subsequent accumulation of MM-CoA and thus causing increased levels of serum methylmalonic acid (MMA) [38] as observed in this study. Our study observed that vitamin $B_{12}$ deficiency was associated with hyperhomocysteinemia. This findings was corroborated by Masoud et al, who observed a high prevalence of vitamin $B_{12}$ deficiency and hyperhomocysteinemia in adults with T2DM in Oman [39]. Similarly, Weikert et al., in their population based prospective study reported the association between reduced serum $B_{12}$ levels and increased serum homocysteine levels [40].

Additionally, participants with vitamin B12 deficiency had higher cardiometabolic risk factors such as high TC and LDL compared to those with normal vitamin B12 levels. This finding is consistent with a study by Al-Daghri et al. who found vitamin B12 levels to be a negative predictor of HDL levels [21]. These highlight the importance of maintaining adequate vitamin $B_{12}$ concentrations in reducing the inflammatory and cardiometabolic risks associated with T2DM.

## Prevalence of peripheral neuropathy

The current study employed one of the best-known methods in determining diabetic neuropathy; the Michigan Screening instrument [27, 41], the MNSI-Q and MNSI-PE and observed diabetic neuropathy of 32.5% and 6.5%, respectively, among our study population. The reported prevalence of diabetic neuropathy is 16.6% in Ghana [42], 29.5% in Ethiopia [43], 56.2% in Yemen [44]. The variations in the prevalence may be due to differences in the types

of diabetes, the different methods of patient selection, the sample size and diagnostic criteria employed. The higher prevalence of neuropathy in this study compared to previous prior study in Ghana was not unexpected since a more sensitive diagnostic instrument was employed in this current study. Furthermore, the duration and dosage of metformin have been identified as independent risk factors for diabetic neuropathy and most of our study population has had diabetes and been treated with metformin for more than 12 years. The study also found vitamin B12 deficiency to be associated with increased risk of neuropathy. This finding is in harmony with a study by Alvarez et al. who found an inverse relationship between plasma level of vitamin B12 and diabetic neuropathy [45]. Other studies have reported similar findings among T2DM patients treated with metformin [46, 47] although there are also conflicting reports [48]. One practical limitation of this study was our inability to perform nerve conduction studies to ascertain whether there was nerve damage among our participants as sometimes, problems with the electrical activity in nerves can cause pain, tingling, or weakness in muscles. We could not also perform nerve biopsy, which has been suggested as gold standard for determination of the presence and types of neuropathy. Therefore, electrophysiological findings could not be confirmed. Another limitation of this study is the relatively small sample size and the cross-sectional nature of the design. The current study design lacked comparator and a control group of healthy subjects, or participants who are not on metformin, this makes it difficult to make causal inference. They are however, useful for establishing preliminary evidence in planning a future advanced study, subsequently, further case-control and prospective studies are warranted.

## Conclusion

Vitamin B12 deficiency and diabetic neuropathy are very high among metformin-treated T2DM patients and it is associated with increased GPA, IFA and TNF-α. Moreover, vitamin B12 deficiency is associated with cardiometabolic risk factors (higher LDL and TC and lower HDL). Upon verification of these findings in a prospective case-control study, it may be beneficial to include periodic measurement of Vitamin B12 using the more sensitive combined indicators (4cB 12) in the management of patients with T2DM treated with metformin in Ghana.

## Supporting information

**S1 File.**
(DOCX)

## Acknowledgments

The authors are grateful to all T2DM participants, the staff of Komfo Anokye Teaching Hospital, Diabetic Clinic for their participation and immense support in making this research a success.

## Author Contributions

**Conceptualization:** Samuel Asamoah Sakyi, Edwin Ferguson Laing, Richard Mantey, Alexander Kwarteng, Eddie-Williams Owiredu, Richard Ephraim Dadzie, Benjamin Amoani, Stephen Opoku, Daniel Boakye.

**Data curation:** Eddie-Williams Owiredu, Richard Ephraim Dadzie, Stephen Opoku, Bright Oppong Afranie.

**Formal analysis:** Alexander Kwarteng, Eddie-Williams Owiredu, Stephen Opoku, Bright Oppong Afranie.

**Funding acquisition:** Samuel Asamoah Sakyi.

**Investigation:** Richard Mantey, Alexander Kwarteng, Stephen Opoku.

**Methodology:** Samuel Asamoah Sakyi, Edwin Ferguson Laing, Richard Mantey, Alexander Kwarteng, Eddie-Williams Owiredu, Richard Ephraim Dadzie, Stephen Opoku, Bright Oppong Afranie.

**Project administration:** Richard Mantey.

**Resources:** Edwin Ferguson Laing, Alexander Kwarteng, Daniel Boakye.

**Supervision:** Samuel Asamoah Sakyi, Edwin Ferguson Laing, Benjamin Amoani, Daniel Boakye.

**Validation:** Samuel Asamoah Sakyi, Eddie-Williams Owiredu, Benjamin Amoani, Bright Oppong Afranie, Daniel Boakye.

**Visualization:** Benjamin Amoani.

**Writing – original draft:** Samuel Asamoah Sakyi, Edwin Ferguson Laing, Richard Mantey, Alexander Kwarteng, Eddie-Williams Owiredu, Richard Ephraim Dadzie, Benjamin Amoani, Bright Oppong Afranie, Daniel Boakye.

**Writing – review & editing:** Samuel Asamoah Sakyi, Edwin Ferguson Laing, Richard Mantey, Alexander Kwarteng, Eddie-Williams Owiredu, Richard Ephraim Dadzie, Benjamin Amoani, Stephen Opoku, Bright Oppong Afranie, Daniel Boakye.

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
