## [Decision Letter · Decision Letter 0]

15 Jan 2021

PONE-D-20-39903

Profiling Immuno-Metabolic Mediators of Vitamin B12 Deficiency Among Metformin-Treated type II Diabetes Patients in Ghana

PLOS ONE

Dear Dr. Sakyi

Thank you for submitting your manuscript to PLOS ONE. After careful consideration, we feel that it has merit but does not fully meet PLOS ONE’s publication criteria as it currently stands. Therefore, we invite you to submit a revised version of the manuscript that addresses the points raised during the review process.

As you can see from the Reviewers' comments below, the principal areas of concern are in the study design and the applicability of the Fedosov methodology

The Reviewers' would want you to include a comparator group consisting of either patients without a diagnosis of type 2 diabetes, or patients with type 2 diabetes but were not treated with metformin.

There is also the need to exclude the diagnosis of type 1 diabetes in the subjects because of high prevalence of other autoimmune conditions in your cohort.

We look forward to receiving your revised manuscript.

Kind regards,

Benjamin Udoka Nwosu, MD

Academic Editor

PLOS ONE

Journal Requirements:

In your Methods section, please provide a justification for the sample size used in your study, including any relevant power calculations (if applicable).

Reviewers' comments:

Reviewer's Responses to Questions

**Comments to the Author**

1. Is the manuscript technically sound, and do the data support the conclusions?

Reviewer #1: No

Reviewer #2: Yes

2. Has the statistical analysis been performed appropriately and rigorously? 

Reviewer #1: Yes

Reviewer #2: Yes

3. Have the authors made all data underlying the findings in their manuscript fully available?

Reviewer #1: Yes

Reviewer #2: Yes

4. Is the manuscript presented in an intelligible fashion and written in standard English?

Reviewer #1: Yes

Reviewer #2: Yes

5. Review Comments to the Author

Reviewer #1: The study addresses an important problem of B12 deficiency associated with metformin use. However, there are many, well conducted prior studies (close to 200) across the globe which have conclusively demonstrated this association. This study is from Ghana, and the authors acknowledge another study from Ghana that demonstrated similar conclusions (This reviewer could not retrieve the citation for this study on Pubmed as referenced by Yakubu at al which showed a lower prevalence of B12 deficiency).

There are several limitations to the study:

1. It is a relatively small study (n=200) and there is no control group.

2. There is a very high prevalence of Vitamin B12 deficiency (40.5%) reported, but in the absence of any control (for example, age/gender matched individuals not on Metformin) it is difficult to attribute this high prevalence only to metformin use. There are numerous other prior studies that are larger, with appropriate controls, which show lower prevalence.

3. In the absence of a control group, a before treatment B12 compared to B12 after treatment with Metformin would provide useful data. This was not done as this is an observational study.

4. The study uses a relatively little used and poorly validated computational model (Fedosov combined marker) to diagnose Vitamin B12 deficiency. Low B12 levels alone are a sensitive indicator of deficiency ( up to 95% sensitive depending on cutoff) (NEJM, 2013. 368:149-60) . In those with borderline low Vitamin B12 (200-300 pg/ml), an elevation of MMA and Homocysteine add to the sensitivity, but both need to be elevated as homocysteine alone can be high in folate deficiency. MMA elevations may not be specific as other conditions including renal disease can cause elevations. Therefore these should be used in situations where Vitamin B12 is borderline low. Holotranscobalmin is neither specific or sensitive but can be useful in pregnancy. There is little data on validity of Fedosov model (Reference 18 in current article) which has been cited only 15 times since its publication in 2013. Reading the original paper reveals this is a theoretical construct which is validated against cognitive impairment in the elderly and hematological impairments.

5. The data correlating B12 deficiency with neurological impairments (two clinical measures of peripheral neuropathy) and immunological parameters does not provide any new information

6. This study shows rates of Vitamin B12 deficiency (40.5%) which is far higher than almost all prior studies, but the non-standard measure (Fedosov wellness quotient) may be the reason for this discrepant finding.

7. Since the average B12 in the group, on page 14, line 287 is reported as 142 pmol/L (which is below the threshold for severe Vit B12 deficiency 148 pmol/L or 200 pg/ml), it would be informative to know what was the percentage of B12 deficiency using just the threshold of 148 pmol/L which is very specific cutoff

8. The study needs to emphasize what new knowledge it adds to the topic of B12 deficiency in Metformin treated patients.

Minor Problems:

Labelling/legends in Figure 1 need to be clearer

Reviewer #2: In the manuscript “Profiling Immuno-Metabolic Mediators of Vitamin B12 Deficiency Among Metformin-Treated type II Diabetes Patients in Ghana” Sakyi et al investigated the immuno-biochemical mediators of vitamin B12 deficiency using Fedosov age normalized wellness quotient. Fedosov proposed the utilization of combined indicator of vitamin B12 status using two or more of the following markers: total serum B12, methylmalonic acid, holo-transcobalamin, and total homocysteine. Comparison of immuno-metabolic parameters based on vitamin B12 status was performed using one-way analysis of covariance (ANCOVA) with adjustment for age, sex, BMI, duration of diabetes, and dosage and duration metformin therapy. The authors report that the prevalence of metformin induced vitamin B12 deficiency was 40.5% whilst the prevalence of diabetic neuropathy was > 67%. Also, Vitamin B12 deficient patients had higher levels of intrinsic factor, gastric parietal cell antibody, TNF-α, total cholesterol and LDL-cholesterol. They conclude that prevalence of Vitamin B12 deficiency and diabetic neuropathy are high among metformin-treated patients with type 2 diabetes and suggest that routine measurement of Vitamin B12 should be included in the management of patients with type 2 diabetes treated with metformin.

Comments:

Strength: a reasonably elaborate study on treatment with metformin and Vitamin B12 deficiency

Major

1. It would be appropriate to clarify how the diagnosis of type 2 diabetes was made. The presence of gastric parietal cell antibody makes it imperative to exclude type 1 diabetes

2. A comparator group such as patients without diabetes or patients with type 2 diabetes but who were not treated with metformin would be required to justify the conclusion that the prevalence of Vitamin B12 deficiency and peripheral neuropathy are high in patients treated with metformin in Ghana. Could there be the possibility that B12 deficiency may be high in the general population.

3. Is there any correlation between the dose/duration of treatment with metformin and Vitamin B12 deficiency/peripheral neuropathy?

4. 2017 ADA guidelines recommended periodic monitoring of Vitamin B12 in patients treated with metformin over a long period of time, suggesting routine measurement in all patients treated with metformin may not be cost effective.

Minor

1. Type II diabetes should read Type 2 diabetes

2. “on metformin” is better rendered as “treated with metformin” throughout the manuscript

3. Figure 4 appears confusing, a table might be easier to understand

6. PLOS authors have the option to publish the peer review history of their article (what does this mean?). If published, this will include your full peer review and any attached files.

Reviewer #1: **Yes: **Samir Malkani

Reviewer #2: No

---

## [Author Response · Author response to Decision Letter 0]

8 Feb 2021

The Editor,

PLOS ONE

Dear Sir/Madam,

RESPONSE TO REVIEW COMMENTS

The authors very much appreciate the timely and scrupulous review of our manuscript (PONE-D-20-39903). Kindly find below the response to the reviewers’ comments. Tracked changes have been employed to highlight texts revised per the reviewers’ recommendations. Revised statements indicated in this response are in “quotation marks”.

Reviewer #1

Comment 1: It is a relatively small study (n=200) and there is no control group.

Response: Thank you for your comment. This observational study is a baseline study to evaluate to use of Fedosov combined marker in Ghanaian population which has not been explored. However, a case-control study with large sample size will add a layer of knowledge. We acknowledged this as a limitation worth exploring by future studies (line 433-4).

“Another limitation of this study is the relatively low sample size and the cross-sectional nature of the design. Further case-control and prospective studies are warranted.”

Comment 2: There is a very high prevalence of Vitamin B12 deficiency (40.5%) reported, but in the absence of any control (for example, age/gender matched individuals not on Metformin) it is difficult to attribute this high prevalence only to metformin use. There are numerous other prior studies that are larger, with appropriate controls, which show lower prevalence.

Response: Thank you for your comment. Although this observational study reported a very high prevalence of vitamin B12 deficiency (40.5%) with the absence of a control group compared to other studies conducted elsewhere, this was not unexpected for reasons such as disparities in environmental and genetic factors that may impact their levels. Additionally, differences in the parameters (MMA, serum B12, etc) considered in determining B12 deficiency is another factor. In an effort to accurately capture patients with true vitamin b12 deficiency, we relied on the combined 4cB12.

Comment 3: In the absence of a control group, a before treatment B12 compared to B12 after treatment with Metformin would provide useful data. This was not done as this is an observational study.

Response: Thank you very much for your comment. The current study was a cross -sectional study where one-point sample was taken. We have highlighted this as a limitation (line 433-4).

Comment 4: The study uses a relatively little used and poorly validated computational model (Fedosov combined marker) to diagnose Vitamin B12 deficiency. Low B12 levels alone are a sensitive indicator of deficiency (up to 95% sensitive depending on cutoff) (NEJM, 2013. 368:149-60). In those with borderline low Vitamin B12 (200-300 pg/ml), an elevation of MMA and Homocysteine add to the sensitivity, but both need to be elevated as homocysteine alone can be high in folate deficiency. MMA elevations may not be specific as other conditions including renal disease can cause elevations. Therefore, these should be used in situations where Vitamin B12 is borderline low. Holotranscobalmin is neither specific nor sensitive but can be useful in pregnancy. There is little data on validity of Fedosov model (Reference 18 in current article) which has been cited only 15 times since its publication in 2013. Reading the original paper reveals this is a theoretical construct which is validated against cognitive impairment in the elderly and hematological impairments.

Response: Thank you for your comment. Although serum B12 presents a good marker for B12 deficiency, several studies have highlighted inconsistencies in using serum B12 to define B12 deficiency. High serum B12 levels can be accompanied by signs of deficiency, and functional deficiency from tissue uptake defects and action of vitamin B12 at the cellular level have been implicated in this association [PMID: 26807790; PMID: 21733877]. There is also evidence that functional vitamin B12 deficiency can occur regardless elevated serum B12 levels and using MMA alone is limited. [PMID: 21733877]. Moreover, elevation of MMA is non-specific in the presence of renal impairment (a somehow common sequela in diabetes) but the use of MMA and homocysteine enhance the sensitivity to identify individuals with borderline vitamin B12 deficiency. Thus, the most accurate marker to true B12 deficiency remains to be identified. However, we believe that a model such as Fedosov 4cB12 which accommodates other markers in addition to serum B12 provides an opportunity.

Comment 5: The data correlating B12 deficiency with neurological impairments (two clinical measures of peripheral neuropathy) and immunological parameters does not provide any new information.

Response: Thank you for your comment. Studies correlating vitamin B12 status to diabetic neuropathy has produced conflicting results [PMID: 27730072; PMID: 28882470; PMID: 27716423]. The two clinical measure of peripheral neuropathy in this study will contribute to the understanding of the relationship between vitamin B12 deficiency is associated and risk of neuropathy.

Comment 6: This study shows rates of Vitamin B12 deficiency (40.5%) which is far higher than almost all prior studies, but the non-standard measure (Fedosov wellness quotient) may be the reason for this discrepant finding.

Response: Thank you very much for your comment. The high prevalence of vitamin B12 deficiency reported in our study (40.5%) is not extremely different from a related study in Ghana which reported high prevalence of B12 deficiency (32.1%) among T2DM patients on metformin [DOI: 10.1080/20905068.2019.1662647]. Additionally, similar to vitamin D deficiency which is very common in the study region despite adequate sunshine, our pilot study found that the use of only serum B12, which is commonly used in other populations overestimates B12 deficiency as evidenced in the higher prevalence of B12 deficiency (>50% of the population) based on serum B12 (Supplementary file: Table S4). In an effort to normalize and accommodate all relevant markers, we used the Fedosov wellness quotient. 

Comment 7: Since the average B12 in the group, on page 14, line 287 is reported as 142 pmol/L (which is below the threshold for severe Vit B12 deficiency 148 pmol/L or 200 pg/ml), it would be informative to know what was the percentage of B12 deficiency using just the threshold of 148 pmol/L which is very specific cutoff.

Response: Thank you for your comment. As indicated in the response to comment 6, the use of serum B12 only in the study setting overestimates the prevalence of B12 deficiency, corroborating previous reports on the disparity between serum B12 levels and functional B12 deficiency (Kindly refer to response to comment 6 and Supplementary file: Table S4).

Comment 8: The study needs to emphasize what new knowledge it adds to the topic of B12 deficiency in Metformin treated patients.

Response: Thank you for your comment. Specifically regarding B12 deficiency in diabetes, as highlighted in line 120-3, the use of individual markers (serum B12, MMA, hTC, Hcy) has rather yielded inadequate reflection of true vitamin B12 status. The study thus highlights the likelihood of missing some cases of B12 deficiency when a single indicator is used compared to using multiple markers (line 120-3 & 378-80).

“Of note, the high rate of vitamin B12 deficiency in this study compared to previous studies could be suggestive that the use of individual markers (serum B12, MMA, hTC or Hcy) underestimates the vitamin B12 status among diabetic populations.”

Minor comment:

Comment 1: Labelling/legends in figure 1 need to be clearer

Response: Thank you for the observation. Labelling/legend in figure 1 has been made clearer. 

Reviewer #2

Major comments:

Comment 1: It would be appropriate to clarify how the diagnosis of type 2 diabetes was made. The presence of gastric parietal cell antibody makes it imperative to exclude type 1 diabetes

Response: Thank you for your observation. The inclusion and exclusion criteria have been clarified (line 177-86)

“The diagnosis of type 2 diabetes mellitus was made based on American Diabetes Association (ADA) criteria (based on defective progressive insulin secretory on the background of insulin resistance using fasting plasma glucose ≥7.0 mmol/L; glycated hemoglobin ≥6.5%; oral glucose tolerance test 11.1 mmol/L)”

“Patients with type 1 diabetes (diagnosed based on high plasma glucose in addition to autoimmune indicators such as autoantibodies to insulin, autoantibodies to glutamate decarboxylase (GAD65)) were excluded.”

Comment 2: A comparator group such as patients without diabetes or patients with type 2 diabetes but who were not treated with metformin would be required to justify the conclusion that the prevalence of Vitamin B12 deficiency and peripheral neuropathy are high in patients treated with metformin in Ghana. Could there be the possibility that B12 deficiency may be high in the general population.

Response: Thank you for your comment. It is possible that there is generally high prevalence of vitamin B12 deficiency in the study population given that a previous study reported over 30% prevalence using serum B12. Additionally, using only serum B12 levels, over 50% of our study population was deficient. Given that this is a baseline study, we have acknowledged the limitation of the cross-sectional design and suggested further studies in this regard (line 433-4).

“Another limitation of this study is the relatively low sample size and the cross-sectional nature of the design. Further case-control and prospective studies are warranted.”

Comment 3: Is there any correlation between the dose/duration of treatment with metformin and Vitamin B12 deficiency/peripheral neuropathy?

Response: Thank you for your comment. Higher dose of metformin (3000mg/day) was associated with increased odds of vitamin B12 deficiency and neuropathy. However, the duration of treatment with metformin was not associated with vitamin B12 deficiency or neuropathy (Supplementary file: Table S2 & S3).

Comment 4: 2017 ADA guidelines recommended periodic monitoring of Vitamin B12 in patients treated with metformin over a long period of time, suggesting routine measurement in all patients treated with metformin may not be cost effective.

Response: Thank you very much for your comment. The sentence has been rephrased.

Minor comment:

Comment 1: Type II diabetes should be read Type 2 diabetes.

Response: Thank you for the comment. ‘Type II diabetes’ has been replaced with ‘Type 2 diabetes’.

Comment 2: “on metformin” is better rendered as “treated with metformin” throughout the manuscript.

Response: Thank you for the suggestion. “on metformin” has been replaced with “treated with metformin” throughout the manuscript. 

Comment 3: Figure 4 appears confusing; a table might be easier to understand.

Response: Thank you very much for your comment. Due to the large number of biochemical and immunological parameters together with the five indicators of vitamin B12, using a table will make it very clumsy. However, further description has been added to make the figure clearer.

Thank you once again for the timely review and scrupulous of our manuscript. Looking forward to hear favorably from you.

Sincerely,

---

## [Decision Letter · Decision Letter 1]

5 Mar 2021

PONE-D-20-39903R1

Profiling Immuno-Metabolic Mediators of Vitamin B12 Deficiency Among Metformin-Treated type II Diabetes Patients in Ghana

PLOS ONE

Dear Dr. Sakyi,

Thank you for submitting your manuscript to PLOS ONE. After careful consideration, we feel that it has merit but does not fully meet PLOS ONE’s publication criteria as it currently stands. Therefore, we invite you to submit a revised version of the manuscript that addresses the points raised during the review process.

Please provide a detailed discussion of the limitation of your study with respect to the lack of a comparator group. This is critical. Your current two-sentence response is not sufficient: 'Another limitation of this study is the relatively low sample size and the cross-sectional nature of the design. Further case-control and prospective studies are warranted.'

You have to discuss the concerns about his lack of comparator group, and what your group should have done in terms of including a control group of healthy subjects, or a group of patients with diabetes but were not receiving metformin.

We look forward to receiving your revised manuscript.

Kind regards,

Benjamin Udoka Nwosu, MD

Academic Editor

PLOS ONE

Journal Requirements:

Additional Editor Comments (if provided):

Please provide a detailed discussion of the limitation of your study with respect to the lack of a comparator group. This is critical. Your current one sentence response, is not sufficient: 'Another limitation of this study is the relatively low sample size and the cross-sectional nature of the design. Further case-control and prospective studies are warranted.'

You have to discuss the concerns about his lack of comparator group, and what your group should have done in terms of including a control group of healthy subjects, or a group of patients with diabetes but were not receiving metformin.

Reviewers' comments:

Reviewer's Responses to Questions

**Comments to the Author**

1. If the authors have adequately addressed your comments raised in a previous round of review and you feel that this manuscript is now acceptable for publication, you may indicate that here to bypass the “Comments to the Author” section, enter your conflict of interest statement in the “Confidential to Editor” section, and submit your "Accept" recommendation.

Reviewer #2: (No Response)

2. Is the manuscript technically sound, and do the data support the conclusions?

Reviewer #2: Yes

3. Has the statistical analysis been performed appropriately and rigorously? 

Reviewer #2: Yes

4. Have the authors made all data underlying the findings in their manuscript fully available?

Reviewer #2: Yes

5. Is the manuscript presented in an intelligible fashion and written in standard English?

Reviewer #2: Yes

6. Review Comments to the Author

Reviewer #2: "It is imperative Ghana include periodic measurement of Vitamin B12 deficiency using the more sensitive combined indicators (4cB 12 ), in the management of T2DM patients treated with metformin" in concluding sentence of the abstract should read "if the findings of this study are verified in a prospective case-control study, it may be beneficial to include periodic measurement of Vitamin B12 using the more sensitive combined indicators (4cB 12 ) in the management of patients with T2DM treated with metformin in Ghana.

7. PLOS authors have the option to publish the peer review history of their article (what does this mean?). If published, this will include your full peer review and any attached files.

Reviewer #2: No

---

## [Author Response · Author response to Decision Letter 1]

8 Mar 2021

The Editor,

PLOS ONE

Dear Sir/Madam,

RESPONSE TO REVIEW COMMENTS R2

The authors very much appreciate review of our manuscript (PONE-D-20-39903). Kindly find below the response to the reviewers’ and editor’s comments. Revised statements indicated in this response are in “highlighted in red”.

Editor’s comment

Please provide a detailed discussion of the limitation of your study with respect to the lack of a comparator group. This is critical. Your current two-sentence response is not sufficient: 'Another limitation of this study is the relatively low sample size and the cross-sectional nature of the design. Further case-control and prospective studies are warranted.'

You have to discuss the concerns about his lack of comparator group, and what your group should have done in terms of including a control group of healthy subjects, or a group of patients with diabetes but were not receiving metformin.

Response: The limitations has been discussed further to read “Another limitation of this study is the relatively small sample size and the cross-sectional nature of the design. The current study design lacked comparator and a control group of healthy subjects, or diabetes who are not on metformin, this makes it difficult to make causal inference. They are however, useful for establishing preliminary evidence in planning a future advanced study, subsequently, further case-control and prospective studies are warranted”. Page 20, line 430-435.

Reviewer #2

Reviewer #2: "It is imperative Ghana include periodic measurement of Vitamin B12 deficiency using the more sensitive combined indicators (4cB 12 ), in the management of T2DM patients treated with metformin" in concluding sentence of the abstract should read "if the findings of this study are verified in a prospective case-control study, it may be beneficial to include periodic measurement of Vitamin B12 using the more sensitive combined indicators (4cB 12 ) in the management of patients with T2DM treated with metformin in Ghana.

Response: The concluding sentence of the abstract and main manuscript has been revised to read " Upon verification of these findings in a prospective case-control study, it may be beneficial to include periodic measurement of Vitamin B12 using the more sensitive combined indicators (4cB 12) in the management of patients with T2DM treated with metformin in Ghana. Page 3, line 78-81

".

Thank you once again for the timely review of our manuscript. Looking forward to hear favorably from you.

Sincerely,

Samuel Asamoah Sakyi

Corresponding author

---

## [Editor Report · Decision Letter 2]

16 Mar 2021

Profiling Immuno-Metabolic Mediators of Vitamin B12 Deficiency Among Metformin-Treated type II Diabetes Patients in Ghana

PONE-D-20-39903R2

Dear Dr. Sakyi,

We’re pleased to inform you that your manuscript has been judged scientifically suitable for publication and will be formally accepted for publication once it meets all outstanding technical requirements.

Kind regards,

Benjamin Udoka Nwosu, MD

Academic Editor

PLOS ONE

Additional Editor Comments (optional):

My comments have been addressed.
---

## [Editor Report · Acceptance letter]

22 Mar 2021

PONE-D-20-39903R2 

Profiling Immuno-Metabolic Mediators of Vitamin B_12_ Deficiency Among Metformin-Treated Type 2 Diabetic Patients in Ghana 

Dear Dr. Sakyi:

I'm pleased to inform you that your manuscript has been deemed suitable for publication in PLOS ONE. Congratulations! Your manuscript is now with our production department. 

Kind regards, 

on behalf of

Dr. Benjamin Udoka Nwosu 

Academic Editor

PLOS ONE